# Antimicrobial resistance among farming communities in Wakiso District, Central Uganda: A knowledge, awareness and practice study

James Muleme[1,2]*, John C. Ssempebwa[1], David Musoke[1], Clovice Kankya[2], Solomon Tsebeni Wafula[1], Justine Okello[2], Lesley Rose Ninsiima[2], Rogers Wambi[2,3], James Natweta Baguma[2], Grace Lubega[1], Brenda Wagaba[1], Sonja Hartnack[4]

1 Department of Disease Control and Environmental Health, School of Public Health, College of Health Sciences, Makerere University, Kampala, Uganda, 2 Department of Biosecurity Ecosystems and Veterinary Public Health, College of Veterinary Medicine Animal Resources and Biosecurity, Makerere University, Kampala, Uganda, 3 Department of Clinical Laboratory Mulago National Referral Hospital P.O, Kampala, Uganda, 4 Vetsuisse Faculty, Section of Veterinary Epidemiology, University of Zurich, Zurich, Switzerland

* mulemej@gmail.com

**Data Availability Statement:** All relevant data are within the manuscript and its Supporting information files.

## Abstract

### Background

Antibiotics are increasingly becoming ineffective as antimicrobial resistance (AMR) continues to develop and spread globally—leading to more difficult to treat infections. Countries such as Uganda are still challenged with implementation of AMR related strategies due to data paucity. This includes a lack of data on the prevailing knowledge and awareness of antimicrobial resistance and antibiotic use among farming communities, both commercial and subsistence, which are instrumental in the implementation of targeted interventions. The aim of our study was to assess the knowledge, attitudes and practices on AMR among subsistence and commercial farmers in Wakiso district, central Uganda.

### Methods

A cross-sectional study was conducted using a semi-structured questionnaire in Wakiso district, Central Uganda in between June and September 2021. Polytomous latent class analyses were performed to group participants based on their responses. Multivariable regression and conditional inference trees were used to determine the association between demographic factors and knowledge on antibiotics and AMR.

### Results

A total of 652 respondents participated in the study among whom 84% were able to correctly describe what antibiotics are. Subsistence farmers (OR = 6.89, 95% CI [3.20; 14.83]), and to a lesser extent, farming community members which obtained their main income by another business (OR = 2.25, 95% CI [1.345; 3.75]) were more likely to be able to describe antibiotics correctly than individuals involved in commercial farming. Based on the latent

**Funding:** This study was funded by Makerere University School of Public Health under the Small Grants Programme, MakSPH-GRCB/18-19/01/02. The funders had no role in study design, data collection and analysis, decision to publish, or preparation of the manuscript.

**Competing interests:** The authors have declared that no competing interests exist.

class analysis, three latent classes indicating different levels of knowledge on AMR, were found. Subsistence farming, higher educational level and younger age were found to be associated with belonging to a class of better knowledge.

## Conclusion

The majority of participants were able to correctly describe antibiotics and aware of AMR, however there was some degree of misunderstanding of several AMR concepts. Targeted AMR interventions should improve awareness and also ensure that not only subsistence farmers, but commercial farmers, are included.

## Introduction

Antimicrobial resistance (AMR) presents an increasing threat to health systems and economies globally [1]. If not well managed, AMR will continue to affect public, animal and environmental health, global trade, agriculture, in addition to undermining a country's effort to attainment of the Sustainable Development Goals (SDGs) [2]. Following global strategies such as the global action plan on AMR, the Global Antimicrobial Surveillance System (GLASS), and the Global Health Security Agenda (GHSA), countries are required to develop and streamline strategies to combat the problem of AMR [3].

In response, Uganda developed for the first time in 2018, a One health AMR National Action Plan and AMR surveillance plan which aims to prevent, slow down, and control the spread of resistant organisms [4]. Despite these efforts, there is still a huge information and data gap regarding the conceptualization of AMR making implementation of such policies and strategies strenuous for Uganda. The current situation is characterized by poor antibiotic stewardship [5], including overuse of antibiotics[6], limited knowledge and awareness on AMR, as well as inadequate surveillance and monitoring of antimicrobial consumption activities, culminating into data paucity limiting a country's capacity to generate implementable strategies for the prevention and control of AMR.

Studies have reported limited awareness about the problem of unscrupulous antimicrobial use, AMR and its consequences to health and economy among farmers [7]. Uganda is characterized with decentralized and privatized (human, veterinary and agrochemical) systems, making regulation a great challenge [8]. The associated risk of AMR has had devastating effects in Uganda especially among communities that practice mixed farming [6]. Uganda has two farming categories of subsistence and commercial. Subsistence farming production is mainly for home consumption and relies on manure as fertilizer and plant herbs to treat their livestock [9]. A large proportion of the Ugandan population is involved in subsistence farming. Commercial farmers pursue production on a large scale while purchasing inputs such as organic fertilizers and antibiotics to treat their livestock as well as growth promotion, which is not (yet) prohibited in Uganda.

In Wakiso district in Central Uganda, densely populated farming communities utilize a vast range of veterinary and agrochemical products such as antibiotics during production [10]. Evidence from a recent Uganda based AMR awareness creation project, it became clear, that there is a paucity of recognisable efforts by Wakiso district authorities, and a dearth of information on knowledge and attitude of farmers on antimicrobial use and development of AMR in Wakiso district [5].

The aim of this study was to assess the proportion of farmers involved in subsistence and commercial farming, as well as members of the farming communities who do farming, but

have another source of income, in Wakiso district, being able to correctly describe an antibiotic. Furthermore, we aimed to assess knowledge and awareness on antibiotic usage and resistance. In addition, the study assessed potential associations between demographic factors and both the ability to correctly describe antibiotics and belonging to different latent classes of knowledge and awareness on antibiotic usage and resistance.

## Material and methods

### Study place, population and design

This study was conducted between June and September 2021 in Wakiso district which is located in the central region of Uganda partly encircling Kampala, Uganda's capital city. The district has an estimated human population of 3 105 700 [10] with annual growth rate of 4.1% and average household size of 3.8. The district is primarily rural with some urban conurbations and some villages in peri-urban settings. The main income generating activities in the area include agriculture, trade, sand mining and stone quarrying. The study population comprised of farming community members, i.e., commercial and subsistence farmers, both crop and animal farming, in addition individuals who did farming, but had another main source of income. The sample size was estimated according to [11]] assuming an alpha of 0.05, a proportion 0.5 being able to correctly describe antibiotics, a standard error of 0.05 and a design effect of 1.5 which leads to a total sample size of 576.

A multi-stage sampling was used. First, six sub counties out of 16 sub counties in Wakiso district were purposively selected due to their high involvement in livestock and crop production (District veterinary office records). Second, two parishes were randomly selected in each sub county, followed by a random selection of two villages per parish. In each village, 27 households were randomly selected following a systematic approach [12]. All households were listed with support from the village chairperson and only those practicing farming were eligible for sampling. At household level, the study involved all farmers (male or female), above 18 years, who had provided consent. Additionally, to be included in the study, they must have used or experienced antimicrobials during their routine livestock production and crop farming. Only households which provided oral consent were included in the study aiming to reach the desired sample size.

A semi-structured interviewer-administered questionnaire was used among farming households. The questionnaire was divided into four sections. The first section (A) comprised questions related to demographic characteristics of the respondent (age, gender, marital status, highest level of education, ethnic group, main source of household income, main occupation of household head, type of farming, main animal species kept, average monthly household income, type of food plants, commonly used medications). At every household, participants were asked if they practice farming for money making (commercial by selling agricultural produce) or farm for home use (subsistence by converting most of the agricultural produce to food and perhaps sell the surplus). Using this criterion, we then classified them as either being subsistence or commercial farmers. The second section (B) contained questions related to knowledge of antibiotics and AMR, including several statements for which the respondents were asked to choose one of the following option "true", "false" or "don't know" to express their (dis-)agreement. Participants were asked to describe what antibiotics are. A proper and correct description was considered if the respondent was able to acknowledge that there are veterinary medicines used at their farm, define their purpose and when they use them, sources of the medicines (such as friends, neighbours, pharmacies, drug shops etc), and give some examples of the commonly used medicines at their farm. Since there is currently no local context of AMR and, or antibiotics in particular, the above criterion was sufficient for this study.

In the third section (C), a Likert scale was used. Respondents were asked to judge—based on a 5-item Likert scale from "strongly disagree" (1) to "strongly agree" (5)—their agreement with eight proposed actions to address the problem of AMR. The fourth section (D) was dedicated to the access and use of antimicrobials in animals. In this section questions related to reared and owned animals, type of drugs used in animal, reasons for using drugs, access to drugs, advice, administration of drugs and waiting times were asked. The questionnaire is presented in the (S1 File).

The questionnaire was pre-tested for its ability to gather adequate and relevant information in Wakiso district in a sub county that was not part of the sampling area. The questionnaires were administered by trained research assistants with diploma and degree levels in a health or social work-related field. The research assistants received a training on ethical conduct, the study subject and the field data collection tool (Kobo Collect) [13]. Data were checked daily in presence of the research assistants and the principal investigator to ensure accuracy and consistence before the next day of data collection.

## Statistical analysis

The statistical analysis was performed with the freely available program R version 4.1.3 [14].

The code for the preparation (S2 File) and the analysis (S3 File), as well as the original data set (S1 Dataset) and the cleaned data set (S4 File) are present in the supplementary material.

**Descriptive analysis.**   For the descriptive analysis, 95% confidence intervals were obtained with the command BinomCI() based Jeffreys approach for binomial data, with MultinomCI() for multinomial data, and with the command MeanCI()for continuous data. All three commands are available in the R package DescTools [15].

**Multivariable regression models and conditional inference trees.**   To assess which demographic variables are significantly associated with the outcome "knowing what antibiotics are", i.e., being able or not to describe antibiotics, we first performed a univariable regression analysis for categorical and continuous predictors. The following variables were included: gender (male or female), marital status (married or not married, including separated, divorced or widowed), education (no school, primary education up to six years, or post-primary education including also a- and o- levels, as well as vocational training), ethnic groups (Baganda, the main tribe in Wakiso district, versus other tribes including Mukiga, Munyankole, Munyoro, Musoga), keeping animals (cattle, goats, sheep, pigs, poultry, rabbits, others) source of income (farming, business or other including both formal and informal employment), income in monthly Ugandan shilling (UGX), type of farming (commercial, subsistence or none) and age in years of the respondent.

All variables (except keeping rabbits) with a resulting p-value below 0.2, were included in a multivariable logistic regression model. For variables with more than two categories, i.e., education, income source and farming type, multiple comparison adjustment was performed with Dunnett's approach based on the R package multcomp [16]. Due to a large number of predictor variables of interest, next to regression models, conditional inference trees, based on recursive partitioning were obtained with the R packages party kit [17, 18] rpart [19] and strucchange [20]. Implausible values for the variable age were imputed with the package missForest [21, 22] assuming MAR (missing at random).

**Polytomous latent class analysis.**   With the aim to detect latent classes in the responses related to asking (dis-)agreement with statements, polytomous latent class analyses were performed separately for the questions with three answer options (B6, B7, B11.1, B11.2, B11.3, B11.4, B11.5, B11.6, B11.7, B11.8 from the original questionnaire recoded into Q1 to Q10) and for the questions with five answer options (C12.1 to C12.8 from the original questionnaire

recoded into C1 to C8). Polytomous latent class analysis was performed with the R package poLCA [23] with 50 000 iterations, 20 estimations starting from 1 to 5 classes. Model selection was based on bic (Bayesian information criterion) and having at least a proportion of 0.1 in each class. With the aim to assess which demographic predictor was significantly associated with class membership, a conditional inference tree was performed.

**Ethical considerations.** The study sought ethical approval from Makerere University School of Public Health Higher Degrees Research and Ethics Committee (SPH-2021-167) and the Uganda National Council for Science and Technology (HS1919ES). In addition, we sought permission to conduct the study from Wakiso district headquarters (Chief administrative officer and district health and veterinary offices). A written informed consent was sought from all the study participants before being recruited into the study. All ethical issues, and confidentiality were followed as guided by the Helsinki declaration [23].

# Results

## Socio-demographic characteristics of the respondents

A total of 652 respondents participated in this study with slightly more commercial farmers (35.4%) and individuals with farming activities, but another main source of income (37.3%) compared to subsistence farmers (27.3%). Amongst the respondents, there were more females (62.9%) and married persons (74.4%). Approximately 39% of the respondents had attained a "post-primary" level of education. Majority of the respondents belonged to the Baganda ethnic group (84%). Most respondents were able to correctly describe antibiotics, i.e., "knowing what antibiotics are" (84%). The most used medications by the respondents and their family were—in decreasing order—antipyretics or painkiller (44.2%), antibiotics (33.7%), other antimicrobials excluding antibiotics (15.2%) and others (6.9%). With respect to drug type, respondents were asked on which ones they see as no longer effective in the treatment of humans and animals. The respondents mentioned antimalarials (39.9%), antipyretics or painkillers (25.5%), antibiotics (17.8%) and other antimicrobials excluding antibiotics (16.9%). Detailed descriptive statistics of demographic data, including the proportions respondents of being able to correctly describe antibiotics, and univariable regression models are presented in Tables 1 and 2.

**Continuous demographic variables and their association with the outcome "knowing what antibiotics" are.** The mean age of the study population was 43 years (95% CI: 42.2–44.4). Individuals which knew what antibiotics are were younger with a mean age of 42.5 years (95% CI: 41.4; 43.6), compared to those not knowing what antibiotics are which had a mean age of 47.5 years (95% CI: 43.6; 51.3), (P = 0.002). Individuals which knew about antibiotics had a higher monthly income (0.47 Mio Ugandan Shilling, 95% CI: 0.377; 0.564) compared to those not knowing about antibiotics (0.215, 95% CI: 0.160; 0.269). More details on species reared, classified per farming type, as well as on reasons to use drugs in animals are presented in Table 2.

Related to questions about knowledge of antibiotics and AMR, more than half of the respondents (52.1%) had antibiotics at home. Respondents were requested to show the research assistants any (human, animal, and crop) based antibiotics. Records of the trade name, route of administration and dosage if available were noted by our study team. The most often shown antibiotics for animals and crops were tetracycline and oxytetracycline. Also shown as "antibiotics" were NSAID ("roket"), vitamins, deworming drugs as well as "yellow powder" and "white liquids". In contrast for humans, the most often presented antibiotics were amoxycilline, ampicilline, cloxacilline, sulfonamid with trimethoprim and ciprofloxacin. Additionally presented as "antibiotics" were NSAIDs, vitamins and antimalarials. When asked, when to stop taking antibiotics, nearly every second respondent said: "when the full

**Table 1. Descriptive statistics for relevant categorical demographic variables and their association with the outcome "knowing what antibiotics" are.**

| Variable | Level | Knowing what antibiotics are | | |
|---|---|---|---|---|
| | | Yes | Total | P-value*[7] |
| | | n (%) [95% CI]*[6] | n (%) [95% CI] | OR*[7] [95%CI] |
| **Gender** | | | | 0.596 |
| | Male | 201 (83.1) [77.9;87.4] | 242 (37.1) [33.5;40.9] | Ref. |
| | Female | 347 (84.6) [80.9;87.9] | 410 (62.9) [59.1;66.5] | 1.12 [0.73;1.73] |
| **Marital status** | | | | <0.001 |
| | Married | 425 (87.6) [84.5;90.3] | 485 (74.4) [70.9;77.6] | Ref. |
| | Not married*[1] | 123 (73.6) [66.6;79.9] | 167 (25.6) [22.4;29.1] | 0.39 [0.25;0.61] |
| **Education** | | | | <0.001 |
| | None | 52 (75.4) | 69 (10.6) [6.6;14.7] | 0.32 [0.16;0.61] |
| | Primary | 196 (77.8) | 252 (38.6) [34.7;42.8] | 0.36 [0.22;0.58] |
| | Post primary*[2] | 300 (90.6) | 331 (50.8) [46.8;54.9] | Ref. |
| **Ethnic group** | | | | 0.002 |
| | Baganda | 472 (86.1) [83.0;88.8] | 548 (84.0) [81.1;86.7] | Ref. |
| | Others*[3] | 76 (73.1) [64.0;80.9] | 104 (16.0) [13.3;18.9] | 0.44 [0.27;0.72] |
| **Income source** | | | | 0.225 |
| | Farming | 336 (82.1) [78.2;85.6] | 409 (62.7) [59.0;66.6] | Ref. |
| | Business | 110 (87.3) [80.7;92.2] | 126 (19.3) [15.6;23.2] | 1.49 [0.85;2.75] |
| | Other*[4] | 102 (87.2) [80.2;92.3] | 117 (17.9) [14.3;21.8] | 1.48 [0.83;2.78] |
| **Farming type** | | | | <0.001 |
| | Commercial | 167 (72.3) [66.3;77.8] | 231 (35.4) [31.3;39.7] | Ref. |
| | Subsistence | 169 (94.9) [91.0;97.5] | 178 (27.3) [23.1;31.6] | 2.62 [1.63;4.21] |
| | Other income | 212 (87.2) [82.6;91.0] | 243 (37.3) [33.1;41.6] | 7.20 [3.47;14.92] |
| **Farming crop or animal** | | | | 0.224 |
| | Animal | 266 (82.1) [77.6;86.0] | 324 (49.7) [45.7;53.8] | Ref. |
| | Crop | 70 (82.35) [73.2;89.3] | 85 (13.0) [9.0;17.2] | 1.02 [0.55;1.96] |
| | None | 212 (87.2) [82.6;91.0] | 243 (37.3) [33.1;41.6] | 1.49 [0.94;2.41] |
| **Main cattle** | | | | 0.493 |
| | No | 370 (83.3) [79.6;86.6] | 444 (68.1) [64.4;71.6] | Ref. |
| | Yes | 178 (85.6) [80.3;89.8] | 208 (31.9) [28.4;35.5] | 1.86 [0.73;1.95] |

*(Continued)*

**Table 1.** (Continued)

| Variable | Level | Knowing what antibiotics are | | | |
|---|---|---|---|---|
| | | Yes | Total | P-value*[7] |
| | | n (%) [95% CI]*[6] | n (%) [95% CI] | OR*[7] [95%CI] |
| **Main goats** | | | | 0.825 |
| | No | 345 (83.7) [79.9;87.0] | 412 (63.2) [59.4;66.8] | Ref. |
| | Yes | 203 (84.6) [79.6;88.7] | 240 (36.8) [33.2;40.6] | 1.09 [0.67;1.70] |
| **Main sheep** | | | | 0.160 |
| | No | 509 (84.7) [81.6;87.4] | 601 (92.2) [89.9;94.0] | Ref. |
| | Yes | 39 (76.5) [63.6;86.4] | 51 (7.8) [5.9;10.1] | 0.59 [0.29;1.28] |
| **Main pigs** | | | | 0.338 |
| | No | 276 (85.4) [81.3;89.0] | 323 (49.5) [45.7;53.4] | Ref. |
| | Yes | 272 (82.7) [78.3;86.5] | 329 (50.5) [46.6;54.3] | 0.81 [0.52;1.26] |
| **Main poultry** | | | | <0.001 |
| | No | 198 (77.9) [72.5;82.7] | 254 (38.9) [35.3;42.7] | Ref. |
| | Yes | 350 (87.9) [84.5;90.7] | 398 (61.0) [57.2;64.7] | 2.06 [1.32;3.22] |
| **Main rabbits** | | | | 0.002 |
| | No | 510 (83.1) [79.9;85.9] | 614 (94.2) [92.2;95.8] | Ref. |
| | Yes | 38 (100) [93.6;100] | 38 (5.8) [4.2;7.8] | Inf. [1.95;Inf] |
| **Main others**[*5] | | | | 0.543 |
| | No | 471 (84.4) [81.2;87.3] | 558 (85.6) [82.7;88.1] | Ref. |
| | Yes | 77 (81.9) [73.2;88.7] | 94 (14.4) [11.9;17.3] | 0.84 [0.46;1.58] |
| **Total** | | 548 (84.0) [81.1;86.7] | | |

*[1] Category "not married" included divorced, separated, widowed and never married.

*[2] Category"post-primary" included A- and O-Level, and technical/vocational training.

*[3] Category "others" included Mukiga, Munyankole, Munyoro and Musoga tribes.

*[4] Category "others" in main income source included also formal and self-employment.

*[5] Main other animals kept included in decreasing order dogs, cats, ducks and rarely turkeys, fish and big rats.

*[6] Binomial confidence intervals following Jeffreys approach.

*[7] p-values and odds ratios present the results from univariable analysis, i.e. Fisher's exact tests or logistic regressions.

dose is over" (49.2%), "when I feel better" (38.5%). Other answers included "when the medicine I bought is half done" (6.7%), "when I get reactions from it" (3.5%) or others (2%). For the following conditions, the respondents thought that they can be treated with antibiotics: cold and flu (55.8%), fever (38.9%), diarrhoea (28.4%), urinary tract infection (17.2%) and aids (6.4%).

**Table 2. Descriptive demographic data on keeping different animal species and reasons to treat animals with antibiotics.**

| Variable | N = 652 | Main source of income | | |
| --- | --- | --- | --- | --- |
| | Total | Other main income | Commercial | Subsistence |
| | n (%) [95% CI] *2 | n (%) [95% CI] | n (%) [95% CI] | n (%) [95% CI] |
| Reared animals (Yes) | 619 (94.9) [93.0;96.0] | 229 (94.2) [90.8;96.6] | 217 (93.9) [90.3;96.5] | 173 (97.2) [93.9;98.9] |
| Own poultry (Yes) | 391 (60.0) [56.2;63.7] | 147 (60.5) [54.2;66.5] | 129 (55.8) [49.4;62.1] | 115 (64.6) [57.4;71.3] |
| Own pigs (Yes) | 314 (48.1) [44.3;52.0] | 104 (42.8) [36.7;49.1] | 118 (51.1) [44.6;57.5] | 92 (51.7) [44.4;58.9] |
| Own cattle (Yes) | 206 (31.6) [28.1;35.2] | 62 (25.5) [20.3;31.3] | 83 (35.9) [29.9;42.3] | 61 (34.3) [27.6;41.4] |
| Own goat-sheep (Yes) | 231 (35.4) [31.8;39.1] | 93 (38.3) [32.3;44.5] | 80 (34.6) [28.7;40.9] | 58 (32.6) [26.0;39.7] |
| Own dog-cat (Yes) | 47 (7.2) [5.4;9.4] | 13 (5.3) [3.0;8.7] | 23 (9.9) [6.6;14.3] | 11 (6.2) [3.3;10.4] |
| **N = 619** | | | | |
| Use drugs to treat sick animals*1 | 434(70.1) [66.4;73.6] | 145 (63.3) [56.9;69.3] | 132 (60.8) [54.2;67.1] | 157 (90.7) [85.7;94.4] |
| Use drugs to prevent animals from becoming sick*1 | 282 (45.5) [41.7;49.5] | 88 (38.4) [33.3;44.8] | 86 (39.6) [33.3;46.2] | 108 (62.4) [55.0;69.4] |
| Use drugs to fatten and, or increase growth of animals*1 | 73 (11.8) [9.4;14.5] | 29 (12.7) [8.8;17.4] | 18 (8.3) [5.2;12.5] | 26 (15.0) [10.3;20.9] |
| Use drugs to kill ticks and other pests*1 | 136 (22.0) [18.8;25.3] | 35 (15.3) [11.1;20.4] | 30 (13.8) [9.7;18.9] | 71 (41.0) [33.9;48.5] |
| Use deworming, loss of appetite, and healing wounds*1 | 38 (6.1) [4.4;8.2] | 12 (5.2) [2.9;8.7] | 18 (8.2) [5.2;12.5] | 8 (4.6) [2.2;8.5] |

*1 Analyzed with a subset of the respondents who rear animals (n = 619).

*2 Binomial confidence intervals following Jeffreys approach.

The most often chosen reasons to normally use drugs in animals were "to treat sick animals" (70.1%), to prevent sickness (45.5%), to fatten and/or increase growth (11.8%), to kill ticks and other pests (22.0%) and for deworming, loss of appetite, healing wounds (6.1%).

Related to questions about access and use of antibiotics, the respondents normally access the antimicrobials for animal use by the veterinary worker (48.6%), the veterinary drug shop (25.2%), by other farmers (12.6%), by the human pharmacy or drug shop (3.5%) or the market (3.5%). When asked why to use human drugs in animals, 17.1% chose at least one, or a combination, of the following reasons: cheaper, easily available, and more effective. Using multiple choice responses (MCRs), 86.4% of the respondents normally got advice on using drugs in animals, most often from the veterinary worker (76.4%), other farmer (24.1%), less often from the package label (5.8%), from human health worker (3.5%) or other capacity (1.8%). Similarly using MCRs, drugs are used following the recommendations from the provider (57%) i.e. (drug shop, veterinarians, medical officers, friends, relatives, neighbours), until the animal is cured (18.2%), until the respondent can afford (12.3%), other reasons (4.4%) or until the package is empty (4%). The veterinary worker (48.9%), followed by the owner of the animal (32.6%), a household member (11.0%), an animal attendant (5.2%) or another person (2.3%) are normally administering the drugs. Asked "do you sell or consume animal products (milk, meat or eggs) from animals that were recently treated with drugs?" about a third of the respondents (35.4%) answered with yes. In decreasing order, the respondents would determine the waiting time: by manufacturer's recommendation (23.6%), by

veterinary advice (21.2%), by own judgement (20.2%), other (8.1%) or advice from human health worker (1%) or no response (26.2%). When asked what is normally done with animal drugs that have expired, empty bottles and sachets, the responses chosen were: burned (51.1%), deposited in communal bins and collected (30.5%), thrown in a pit latrine (16.1%), placed in a rubbish pit next to house (14.4%), placed in a communal rubbish pit (12.3%), other (9.4%), thrown in a drain/open area (6.8%). Three implausible values for age were imputed.

## Outcome: Knowing what antibiotics are

Based on the results from the univariable analysis, the following variables—with p < 0.2— were included in the multivariable regression model: marital status, education, farming, ethnic group, keeping poultry, age, and income (Table 3). Subsistence farmers (OR = 6.89, 95% CI [3.20; 14.83]), and to a lesser extent, individuals who have other source of income other than farming (OR = 2.25, 95% CI [1.345; 3.75]) were more likely to be able to describe antibiotics correctly compared to individuals involved in commercial farming. Individuals who were married were ~1.9 times (OR = 1.90, 95% CI [1.16; 3.08]) more likely to know what antibiotics are, compared to those that were not married. Baganda were 2 times more likely to know what antibiotics are (OR = 2.25, 95% CI [1.28; 3.93]) than those belonging to other ethnic groups. Participants who kept birds were ~ 2 times more likely to know what antibiotics are (OR = 1.83, 95% CI [1.14; 2.92]) compared to those that did not keep birds. The variable keeping rabbits was excluded to avoid convergence problems as all respondents keeping rabbits (38/652) knew about antibiotics. Younger individuals and individuals with higher incomes were more likely to know about antibiotics.

**Conditional inference trees.** Similar to the regression models, farming was the variable being most closely associated with knowing what antibiotics are. Individuals practicing commercial farming with no formal or just primary education showed the lowest level of

**Table 3. Results of the multivariable logistic regression to assess potential associations with of the outcome variable "knowing what antibiotics are" with demographic factors.**

| Variable | Level | Effect size (OR) | [95% CI] | p-value |
|---|---|---|---|---|
| **Marital status (Ref = not married)** | | 1.90 | [1.16;3.08] | 0.010 |
| **Education** | | | | 0.186 |
| | Post | | Ref. | |
| | Primary | 0.61 | [0.36;1.04] | |
| | None | 0.68 | [0.31;1.52] | |
| **Farming** | | | | <0.001 |
| | Commercial | | Ref. | |
| | Subsistence | 6.89 | [3.20;14.83] | |
| | Other source of income | 2.25 | [1.35;3.75] | |
| **Ethnic group Baganda (Ref = others)** | | 2.25 | [1.28;3.93] | 0.004 |
| **Keep birds (Ref = no)** | | 1.83 | [1.14;2.92] | 0.011 |
| **Age in years** | | 0.98 | [0.97;0.99] | 0.029 |
| **Monthly income[*1]** | | 1.0000014 | [1.0000005; 1.0000025] | 0.009 |

[*1] In Millions of Ugandan Shillings (UGX).

knowledge about antibiotics. Amongst subsistence farmers and individuals with another source of income than farming activities, age was found to be significant: older individuals were less likely to be able to describe what antibiotics are. Amongst individuals younger than 57 years, subsistence farmers showed a higher level of knowledge on antibiotics when compared to farming individuals with another source of main income. A conditional inference tree including all relevant predictors for the outcome "knowing what antibiotics are" is presented in Fig 1. In contrast to the regression models, marital status, income, tribe and keeping birds were no longer significantly associated with knowledge on antibiotics.

**Questions related to knowledge and attitudes towards antibiotics.** The proportions and 95% binomial confidence intervals of the correct, wrong and unknown answers for the 10 questions (Q1 to Q10) classified per farming type are presented in Table 4. For most of the questions, subsistence farmers chose more often the correct answers than commercial farmers and health professionals/not involved in farming. For the first two questions related to a restricted use of antibiotics, at least 60% of the subsistence farmers chose the correct option, compared to a considerably lower proportion of correct answers in health personnel and commercial farmers. The third question related to antibiotic resistance was answered mostly wrong. While the majority (approximately 60 to 80%) of the respondents were aware of the increasing risk and negative consequences due to AMR, including the possibility that the

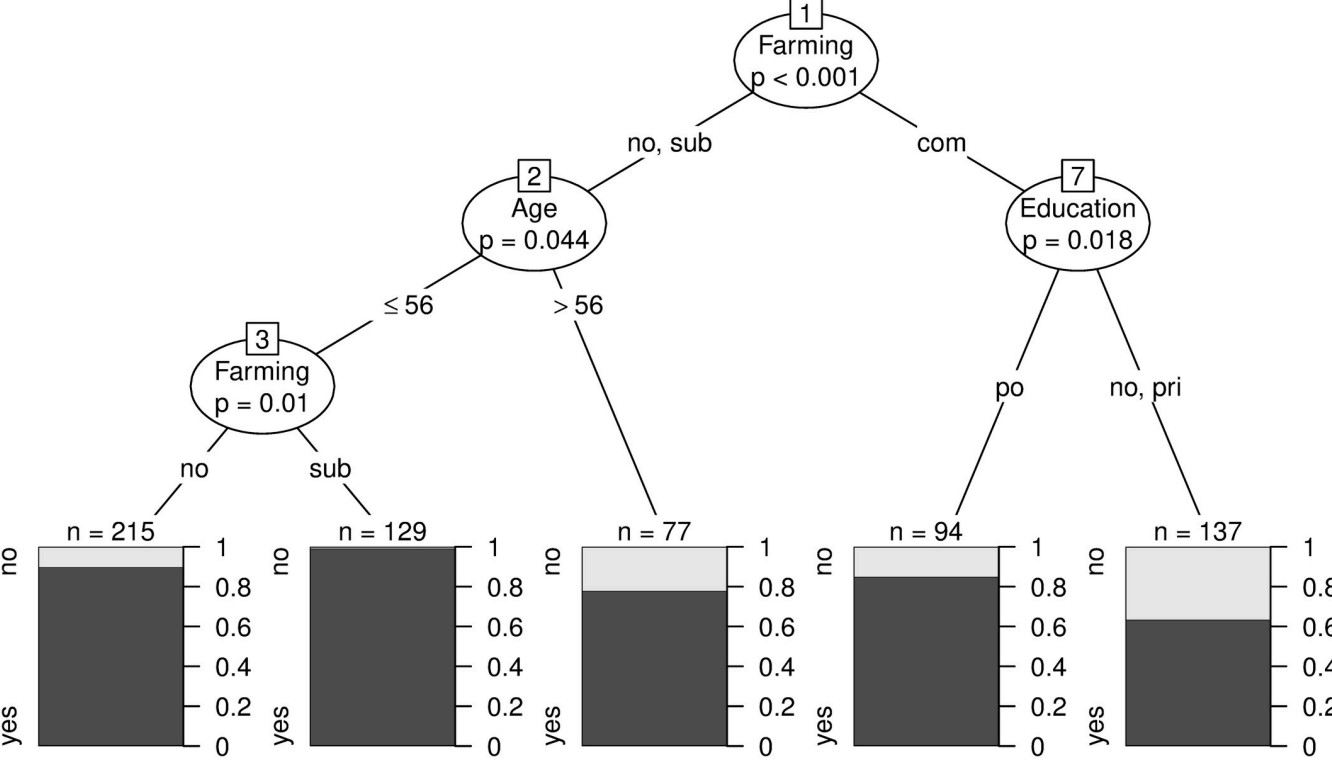

**Fig 1. Conditional inference tree for the outcome "knowing what antibiotics are" including all potential predictor variables.** The following variables were included in the conditional inference tree: Age, gender, marital status, education, ethnic group, main source of household income, income in UGX, farming type, and keeping as main species cattle, goats, sheep, pigs, poultry, others. Farming types are abbreviated with "com" (commercial), "sub" (subsistence) and "no" (other source of main income as farming). Education levels are abbreviated with "no" (nonformal), ""pri" (primary) and "post" (post-secondary). The most closely associated variable with the outcome variable is farming. Amongst subsistence farmer and health professionals, the next relevant variable is age. Individuals younger than 57 years are more likely to know what antibiotics are compared to older respondents. Among younger individuals, subsistence farmer did more often know what antibiotics are, compared to health professionals not involved in farming. Amongst commercial farmers, the level of education was relevant. Commercial farmers with post-primary education were more likely to know what antibiotics are.

**Table 4. Proportions of correct, wrong or "don't know" chosen options classified by type of farming for ten statements related to knowledge of antibiotics and AMR.**

| Questions | Option | Farming type | | | |
|---|---|---|---|---|---|
| | | Other main income n (%) [95%CI] | Commercial n (%) [95%CI] | Subsistence n (%) [95%CI] | Total n (%) [95%CI] |
| Q1.* It's okay to use antibiotics that were given to a friend or family member, as long as they were used to treat the same illness. Correct option: "No". | | | | | |
| | Correct | 115 (47.3) [41.1;54.2] | 123 (53.2) [46.7;60.0] | 120 (67.4) [60.7;74.4] | 358 (54.9) [51.1;59.0] |
| | Wrong | 120 (49.4) [43.2;56.2] | 96 (41.5) [35.1;48.4] | 55 (30.9) [24.1;37.9] | 271 (41.6) [37.7;45.7] |
| | Unknown | 8 (3.3) [0;10.1] | 12 (5.2) [0;12.0] | 3 (1.7) [0;8.7] | 23 (3.5) [0;7.6] |
| Q2.* It's okay to buy the same antibiotics, or request these from a doctor, if you're sick and they helped you get better when you had the same symptoms before. Correct option: "No" | | | | | |
| | Correct | 81 (33.3) [27.6;39.7] | 98 (42.2) [35.9;49.1] | 110 (61.8) [55.0;69.5] | 289 (44.3) [40.5;48.4] |
| | Wrong | 159 (65.4) [59.7;71.7] | 126 (54.4) [48.0;61.3] | 67 (37.6) [30.9;45.3] | 352 (53.9) [50.1;58.1] |
| | Unknown | 3 (1.2) [0;7.6] | 7 (3.0) [0;9.7] | 1 (0.6) [0;8.3] | 11 (1.7) [0;5.8] |
| Q3.* Antibiotic resistance occurs when your body becomes resistant to antibiotics and they no longer work as well. Correct option: "No" | | | | | |
| | Correct | 18 (7.4) [3.3;11.7] | 17 (7.4) [1.7;13.3] | 25 (14.0) [9.0;19.7] | 60 (9.2) [6.3;12.3] |
| | Wrong | 205 (84.4) [80.2;88.7] | 163 (70.5) [64.9;76.5] | 145 (81.5) [76.4;87.1] | 513 (78.7) [75.8;81.8] |
| | Unknown | 20 (8.2) [4.1;12.6] | 51 (22.1) [16.4;28.0] | 8 (4.5) [0;10.2] | 79 (12.1) [9.2;15.2] |
| Q4.* Many infections are becoming increasingly resistant to treatment by antibiotics. Correct option: "Yes" | | | | | |
| | Correct | 180 (74.1) [69.1;79.7] | 181 (78.3) [73.6;83.7] | 141 (79.2) [73.6;84.9] | 502 (77.0) [73.9;80.2] |
| | Wrong | 34 (14.0) [9.0;19.7] | 15 (6.5) [1.7;11.9] | 29 (16.3) [10.7;22.0] | 78 (12.0) [8.9;15.1] |
| | Unknown | 29 (11.9) [7.0;17.6] | 35 (15.1) [10.4;20.5] | 8 (4.5) [0;10.2] | 72 (11.0) [8.0;14.2] |
| Q5.* If bacteria are resistant to antibiotics, it can be very difficult or impossible to treat the infections the cause. Correct option: "Yes" | | | | | |
| | Correct | 197 (81.1) [76.5;85.8] | 159 (68.8) [63.2;74.9] | 148 (83.1) [78.1;88.3] | 504 (77.3) [74.2;80.5] |
| | Wrong | 29 (11.9) [7.4;16.7] | 43 (18.6) [13.0;24.7] | 21 (11.8) [6.7;16.9] | 93 (14.3) [11.2;17.4] |
| | Unknown | 17 (7.0) [2.5;11.7] | 29 (12.5) [6.9;18.7] | 9 (5.0) [0;10.2] | 55 (8.4) [5.4;11.6] |
| Q6.* Antibiotic resistance is an issue that could affect me or my family. Correct option: "Yes" | | | | | |
| | Correct | 203 (83.5) [79.4;88.2] | 197 (85.3) [81.4;89.8] | 143 (80.3) [75.3;86.3] | 543 (83.3) [80.7;86.1] |
| | Wrong | 29 (11.9) [78.2;16.6] | 22 (9.5) [5.6;14.1] | 27 (15.2) [10.1;21.1] | 78 (12.0) [9.3;14.8] |
| | Unknown | 11 (4.5) [0.4;9.2] | 12 (5.2) [1.3;9.7] | 8 (5.2) [1.2;9.7] | 31 (4.7) [2.1;7.5] |
| Q7.* Antibiotic resistance is an issue in other countries, but not in our country. Correct option: "No" | | | | | |
| | Correct | 97 (39.9) [33.3;46.6] | 136 (58.9) [52.8;65.7] | 110 (61.8) [55.0;69.4] | 343 (52.6) [48.6;56.7] |
| | Wrong | 110 (45.3) [38.7;52.0] | 52 (22.5) [16.4;29.3] | 46 (25.8) [19.1;33.4] | 208 (31.9) [27.9;36.0] |

(*Continued*)

**Table 4.** (Continued)

| Questions | Option | Farming type | | | |
|---|---|---|---|---|---|
| | | Other main income n (%) [95%CI] | Commercial n (%) [95%CI] | Subsistence n (%) [95%CI] | Total n (%) [95%CI] |
| | Unknown | 36 (14.8) [8.2;21.5] | 43 (18.6) [12.5;25.5] | 22 (12.3) [5.6;19.9] | 101 (15.5) [11.5;19.6] |
| Q8.* Antibiotic resistance is only a problem for people who take antibiotics regularly. Correct option: "No" | | | | | |
| | Correct | 56 (23.0) [17.3;29.1] | 66 (28.6) [22.1;35.2] | 32 (18.0) [11.8;24.9] | 154 (23.6) [19.9;27.4] |
| | Wrong | 161 (66.2) [60.5;72.3] | 130 (56.3) [49.8;62.9] | 125 (70.2) [64.0;77.1] | 416 (63.8) [60.1;67.6] |
| | Unknown | 26 (10.7) [4.9;16.8] | 35 (15.1) [4.9;16.8] | 21 (11.8) [5.6;18.7] | 82 (12.6) [8.9;16.4] |
| Q9.* Bacteria which are resistant can spread from person to person. Correct option: "Yes" | | | | | |
| | Correct | 203 (83.5) [79.4;88.2] | 172 (74.5) [69.3;80.1] | 134 (75.3) [69.7;81.8] | 509 (78.1) [75.0;81.1] |
| | Wrong | 29 (11.9) [7.8;16.6] | 32 (13.8) [8.6;19.5] | 29 (16.3) [10.7;22.8] | 90 (13.8) [10.7;16.9] |
| | Unknown | 11 (4.5) [0.4;9.2] | 27 (11.7) [6.5;17.3] | 15 (8.4) [2.8;14.9] | 53 (8.1) [5.1;11.2] |
| Q10.* Antibiotic-resistant infections could make medical procedures like surgery, organ transplants and cancer treatment much more dangerous Correct option: "Yes" | | | | | |
| | Correct | 168 (69.1) [63.8;75.2] | 117 (50.6) [44.1;57.7] | 110 (61.8) [55.0;69.4] | 395 (60.6) [56.7;64.4] |
| | Wrong | 39 (16.0) [10.7;22.1] | 46 (19.9) [13.4;27.0] | 29 (16.3) [9.5;23.9] | 114 (17.5) [13.6;21.3] |
| | Unknown | 36 (14.8) [9.5;20.8] | 68 (29.4) [22.9;36.5] | 39 (21.9) [15.2;29.5] | 143 (21.9) [18.1;25.8] |

* Q1 to Q10 correspond to B6, B7, B11.1, B11.2, B11.3, B11.4, B11.5, B11.6, B11.7, B11.7, B11.8.

respondent or his/her family could be affected, lower proportions considered AMR to be an issue in Uganda compared to other countries. Most respondents agreed (wrongly) with the statement, that AMR is only a problem for people who take antibiotics regularly. A clear majority agreed correctly with the statement that resistant bacteria can spread from person to person. To a lesser extent, the participants agreed that medical procedures could become more dangerous due to antibiotic-resistant infections.

Based on the polytomous latent class analysis, the data was best described with three latent classes which we named "appropriate knowledge" (Class 1, 40.3%), "partially wrong" (Class 2, 40.3%), and "limited knowledge" (Class 3, 36.8%) (Fig 2). Individuals in class 1 were most likely to respond correctly to all questions with the exception of Q3, Q8 and Q9. Individuals in class 2 chose mostly or often the wrong answers for Q1, Q2, Q3, Q7 and Q8, but the correct answers for Q4, Q5, Q6, and Q10. Individuals to class 3 showed more variability in their answer pattern with Q6 being mostly answered correctly. Individuals from class 3 chose also relatively often the option "I don't know".

Similarly, to the previous regression models for the outcome knowing what antibiotics are, farming type was the variable being most closely associated with class membership in a conditional inference tree (Fig 3). Subsistence farmers were more likely to be in class 1 compared to commercial and individuals with another main source of income as farming activities. Commercial farmers or no farmers were more likely to belong to class 1 when they had a higher level of education, compared to a lower level of education, but still a high proportion belonged to a

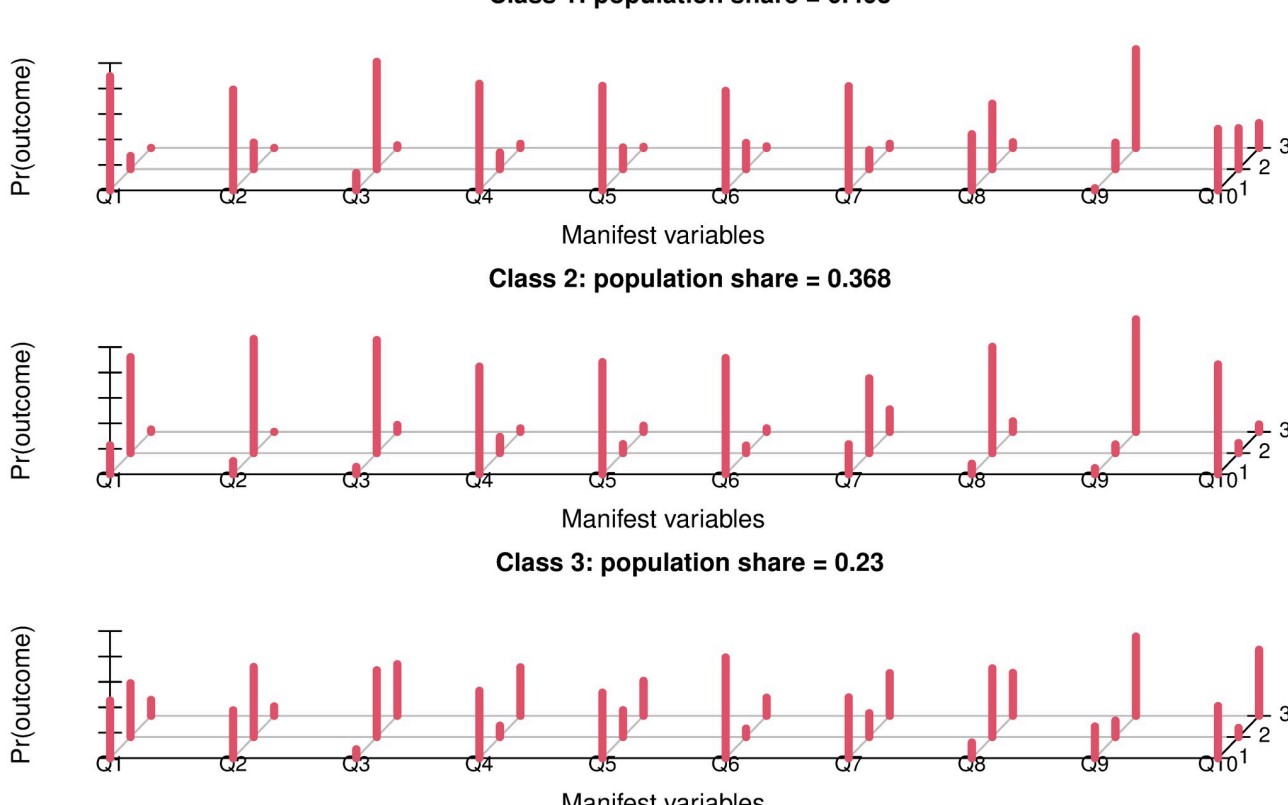

**Fig 2. Results of the polytomous latent class analysis analysing responses to 10 statements related to knowledge and attitudes towards antibiotics.**
The data are best described by three latent classes, which we named "appropriate knowledge" (class 1), "partially wrong" (class 2) and "limited knowledge" (class 3). The x-axes indicate each of the ten questions, the y-axes the frequencies of the three options of correct, wrong, or don't know options which are indicated with 1, 2, and 3 in the z-axis (Fig 2).

class with partial knowledge on antibiotics and AMR. Commercial farmer with no post-primary education belonged most likely to a class with a limited knowledge on antibiotics and AMR.

**Questions related to proposed actions to address AMR.** The proportions and 95% multinomial confidence intervals of the five options ranging from 1 ("disagree strongly") to 5 ("agree strongly") for the questions C1 to C8, classified per farming type are presented in Table 5. For all eight questions, a clear majority agreed with the eight statements. For most of the statements, subsistence farmers scored highest.

Based on a polytomous latent class analysis, two classes were found. One statement was excluded, since only four out of five possible answer options were chosen. We named the first class "definitive agreement"—class 1 (64.7%) and the second class "mixed agreement"- class 2 (35.3%) (Fig 4).

In a conditional inference tree, to assess the association between all relevant demographic predictor variables and class membership, age was the only variable closely related to class membership (S1 Fig). Elderly participants were more likely to agree strongly with the statements.

## Discussion

This study was conducted among 652 respondents, with backgrounds in subsistence and commercial farming, as well as those whose main source of income was not farming. The study

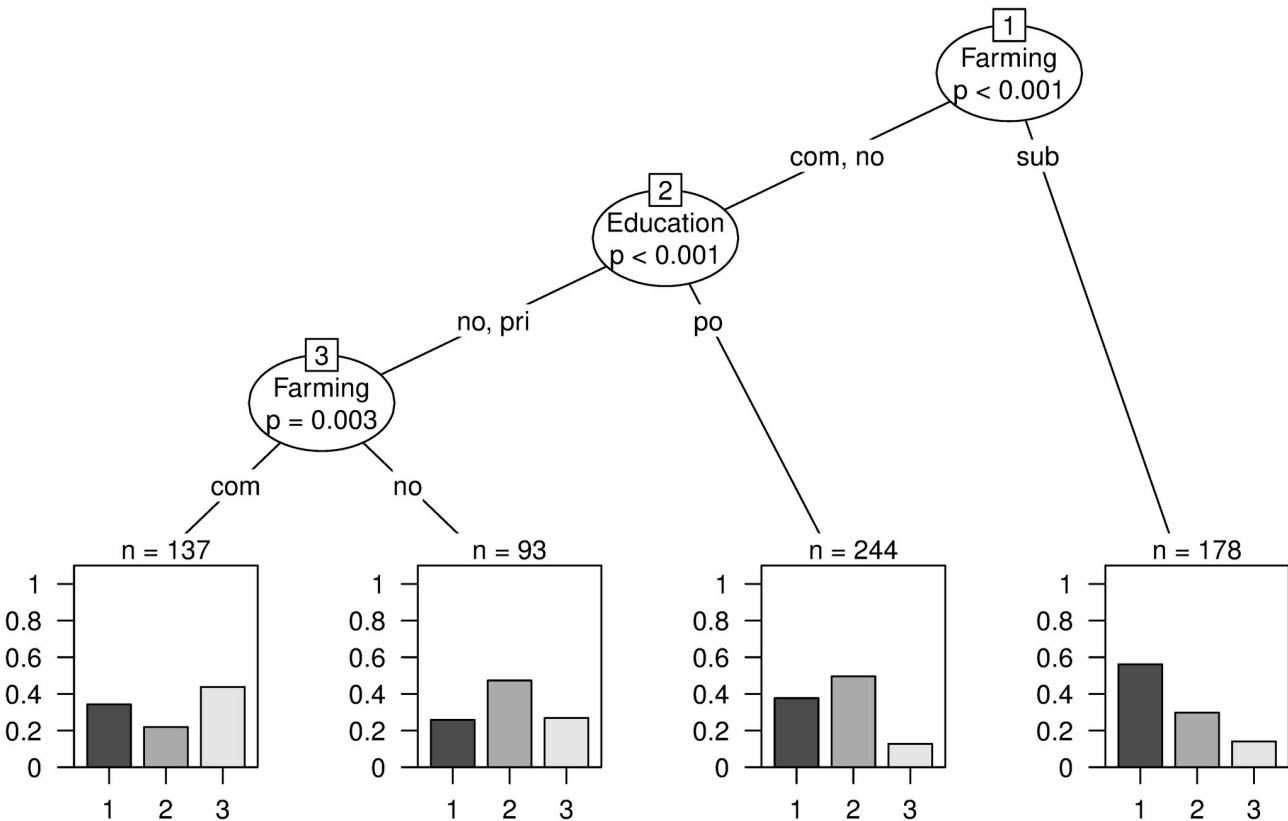

**Fig 3. Conditional inference tree for the outcome class membership including all potential predictor variables.** The farming types are abbreviated with "sub" for subsistence, "no" for not farming as a main source of income and "com" for commercial farming. Education levels are indicated with "no, pri" for no and primary education, "po" for post-primary education. Based on the polytomous latent class analysis, class 1 designated "appropriate", class 2 "partially wrong" and class 3 "limited knowledge" on antibiotics and AMR. Compared to commercial farmers and health professionals not involved in farming, amongst the subsistence farmers was the highest proportion of individuals belonging to class 1, the highest level of antibiotic and AMR knowledge (Fig 3).

took place in Wakiso district, an area which is densely populated by humans and their animals. Independent of their farming background, most of the respondents kept animals.

A large proportion could describe antibiotics correctly, i.e., knew what antibiotics are. The overall level of knowledge in this study regarding the AMR was found to be 80% which was slightly higher than in a study which reported about 65.2% knowledge levels regarding antibiotic use [24]. This contrasts with a study done among pig farmers indicating that knowledge of antibiotics was very poor [25]. There is need for researchers and internationally recognized research bodies to generate a knowledge measurement scale to enable a proper comparison of knowledge across different studies and settings.

Subsistence farming, higher level of education and younger age were associated with a better knowledge on antibiotics. Next to classical multivariable regression analysis, which might be affected by a potential overfitting [26], we used also conditional inference trees, a non-parametric approach which resulted in similar findings. Majority of individuals practicing subsistence were able to answer questions related to knowledge of antibiotics and AMR more correctly compared to individuals with another main source of income than farming and commercial farmers. Farming type was the variable being closest associated with the variable "knowing what antibiotics are" and latent class membership. Subsistence farmers were more

**Table 5. Proportions of correct, wrong or "don't know" chosen options classified by type of farming for eight statements related to proposed actions to address AMR.**

| Question | | | | | | Farming | | |
|---|---|---|---|---|---|---|---|---|
| | Option 1*[2] | Option 2 | Option 3 | Option 4 | Option 5 | None | Commercial | Subsistence |
| | n (%) [95%CI] *[3] | n (%) [95%CI] | n (%) [95%CI] | n (%) [95%CI] | n (%) [95%CI] | mean (sd) | mean (sd)*[4] | mean (sd) |
| C1. People should use antibiotics only when they are prescribed by a health practitioner. | | | | | | | | |
| | 114 (17.5) [14.4;20.7] | 23 (3.5) [0.5;6.7] | 10 (1.5) [0;4.7] | 1 (0.1) [0;3.4] | 504 (77.3) [74.2;80.5] | 3.89 (1.7) | 4.25 (1.47) | 4.41 (1.39) |
| C2. Farmer should give fewer antibiotics to food producing animals. | | | | | | | | |
| | 162 (24.8) [21.2;28.7] | 33 (5.1) [1.4;9.0] | 42 (6.4) [2.8;10.3] | 19 (2.9) [0;6.8] | 396 (60.7) [57.0;64.6] | 3.35 (1.8) | 3.56 (1.8) | 4.34 (1.4) |
| C3. People should not keep antibiotics and use them later for other illnesses. | | | | | | | | |
| | 122 (18.7) [15.2;22.5] | 58 (8.9) [5.4;12.7] | 46 (7.0) [3.5;10.8] | 11 (1.7) [0;5.5] | 415 (63.6) [60.1;67.4] | 3.53 (1.7) | 3.76 (1.6) | 4.31 (1.5) |
| C4. Parents should make sure all of their children's vaccinations are up to date | | | | | | | | |
| | 96 (14.7) [11.6;17.7] | 13 (2.0) [0;5.1] | 7 (1.1) [0;4.2] | 32 (4.9) [1.8;8.0] | 504 (77.3) [74.2;80.4] | 4.18 (1.6) | 4.28 (1.4) | 4.41 (1.4) |
| C5. People should wash their hands regularly. *[1] | | | | | | | | |
| | 52 (8.0) [6.0;10.0] | 3 (0.5) [0;2.5] | 1 (0.1) [0;2.2] | 596 (91.4) [89.4;93.5] | 0 (0) [0;2.0] | 3.78 (0.8) | 3.77 (0.8) | 3.7 (0.9) |
| C6. Doctors should only prescribe antibiotics when they are needed | | | | | | | | |
| | 111 (17.0) [13.6;20.6] | 46 (7.0) [3.7;10.7] | 42 (6.4) [3.1;10.1] | 8 (1.2) [0;4.8] | 445 (68.2) [64.9;71.9] | 3.86 (1.6) | 3.83 (1.6) | 4.29 (1.5) |
| C7. Governments should reward the development of new antibiotics. | | | | | | | | |
| | 156 (23.9) [20.5;27.4] | 8 (1.2) [0;4.7] | 4 (0.6) [0;4.1] | 10 (1.5) [0;5.0] | 474 (72.7) [69.3;76.2] | 3.83 (1.8) | 4.04 (1.7) | 4.11 (1.6) |
| C8. Pharmaceutical companies should develop new antibiotics | | | | | | | | |
| | 156 (23.9) [20.5;27.4] | 9 (1.4) [0;0.5] | 7 (1.1) [0;4.5] | 6 (0.9) [0;4.4] | 474 (72.7) [69.3;76.2] | 3.74 (1.8) | 4.12 (1.6) | 4.09 (1.7) |

*[1]Statement C5 was not included in the polytomous latent class analysis as there were only four levels.

*[2] The answer options included 1 ("strongly disagree") to 5 ("strongly agree").

*[3] Multinomial confidence intervals.

*[4] Standard deviation.

likely to be in class 1- designating the class with the highest proportion of correct answers—compared to commercial farmers and individuals without farming activities. Among the subsistence farmers, the younger participants showed higher level of agreement with the statements related to proposed actions to address AMR, hence high level of awareness. Contrary, a study done among young subsistence farmers in Jordan, indicated unsatisfactory knowledge and awareness levels about proper antibiotic use [27]. In addition, since subsistence farmers do agriculture on a small scale, they would not want to make much loses. Therefore, they may seek for knowledge and advice regarding the use of antibiotics and other chemical substances to obtain high yield. Interventions should target commercial farmers to ensure sustainable behaviour change and increased awareness.

Interestingly, individuals which kept poultry as main species showed the highest level of knowledge on antibiotics. This might reflect sensitisation and awareness efforts in this sector.

Gender was never found to be significantly associated with knowledge on antibiotics. More than half of the respondents had antibiotics at home.

**Class 1: population share = 0.647**

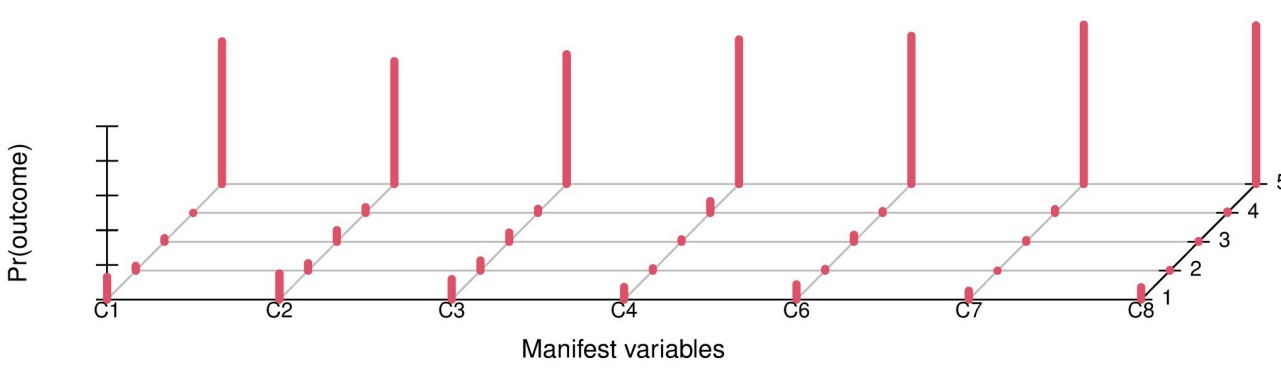

**Class 2: population share = 0.353**

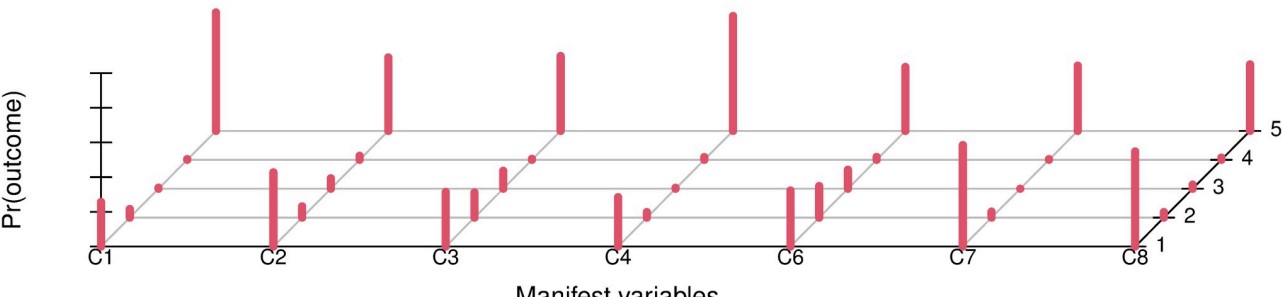

**Fig 4. Results of the polytomous latent class analysis analysing responses to 7 statements related to proposed actions to address AMR.** The data were best described with two latent classes which we named "definitive agreement" (class 1, 64.7%) and "mixed agreement" (class2, 35.3%). In the x-axis the seven statements are indicated, in the y-axis the frequencies of the 5 chosen options based on a Likert scale from 1 ("strongly disagree") to 5 ("strongly agree") which are presented on the z-axis (Fig 4).

Still, based on descriptions like "yellow powder" or "white liquids", it became evident, that the respondents were not always able to correctly classify a drug as an antibiotic.

Respondents were aware that antibiotics may become less effective, but the highest proportion of drugs which are no longer effective, were antimalarials. It is well possible, that resistance is also increasing in antimalarials, but it is also possible that the underlying disease condition was not malaria, but a fever of another origin.

Although a majority agreed that fewer antibiotics should be given to animals, that antibiotics should only be taken after prescription by a doctor, and an awareness of AMR and its detrimental effects was detectable, a number of answers and agreed statements gave evidence of an attitude which favours the further development of AMR. More than a third of the respondents is willing to stop an antibiotic treatment earlier when they feel better. Worrisome is also, that nearly one in five respondents consider using human antibiotics in animals because they are cheaper, easily available and more effective. The unsafe cross-over use of antibiotics has also been reported elsewhere [16].

About a third stated also that they sell or consume milk, meat and eggs from animals treated recently with antibiotics. This is lower compared to a study in Kampala where more than 90% of the commercial egg producer do not observe withdrawal periods [27]. Improper drug usage and failure to keep the withdrawal period has been described as the most important reason for the presence of drug residues [28]. Relevant is also the disposal of expired drugs, empty bottles

and sachets which are a number of times dealt with inappropriately risking the release of antibiotics into the environment.

Related to the agreements with specific statements, following the approach from [29], we realised that one question (Q3) might have been phrased in ambiguous way. It might be understood as two questions. Another possibly ambiguous term might be the statement, that AMR is an issue in other countries, but not in Uganda, while the other responses gave evidence that the respondents are aware of AMR. Possibly the respondents understood that AMR is not an issue in the society or on the political agenda in Uganda.

A misunderstanding, which should be clarified in future intervention campaigns, is that AMR is just an issue for people taking antibiotics regularly. This misunderstanding is particularly relevant in Uganda, lacking a systematic surveillance of residues in milk, meat and eggs and a waste management system being able to separate different types of waste [30]. Positive is the high level of agreement with general hygiene practices.

To some extent, the lack of a clear local nomenclature of AMR and, or antimicrobials agents possibly introduced some degree of interviewer bias which should be considered as a study limitation. However, we aimed to control for this by asking more questions regarding antibiotics, applying latent class analysis to cluster respondents' answers as well as asking the respondents to show evidence for the drugs they have in their house. Both analyses, i.e., focussing on the outcome "being able to correctly describe what antibiotics are" as well as focussing on questions related to knowledge and attitudes towards antibiotics, indicated that commercial farmers had a lower knowledge compared to subsistence farmer and individuals with another source of main income than farming. Potentially, since commercial farmers hire specialised labour from veterinarians, they take little or no time to think about the use of these antibiotics thus reluctant to seek knowledge and information on the same. Responses obtained in our study might have been affected by the earlier studies and interventions therefore not suitable for generalisation.

## Conclusions

Results from both the multivariable regression models, and the polytomous latent class analysis revealed that most participants were able to correctly describe antibiotics and aware of AMR, but misunderstanding is still present. Targeted AMR interventions should assure that not only subsistence farmers, but commercial farmers, are included.

## Supporting information

**S1 Fig. Conditional inference tree to assess class membership of the polytomous latent class analysis based on the statements related to proposed actions to address AMR.**
(TIF)

**S1 Data set. This is the raw data set as collected which can be analysed with the code in S2 and S3.** The variable names correspond to the questions in the questionnaire in S1.
(CSV)

**S1 File. Questionnaire.** This questionnaire was administered by interviewer in English or Luganda.
(PDF)

**S2 File. Code for data preparation.** This reproducible R code was used to prepare the raw data set for the statistical analysis.
(RNW)

**S3 File. Code for statistical analysis.** This reproducible R code was used for the statistical analysis and generating the figures.
(RNW)

**S4 File. Cleaned data set.** This is the cleaned data set which was prepared by applying code in S2 and then used with code in S3.
(R)

## Acknowledgments

We would like to thank the farmers in Wakiso district, the district officials (chief administrative officer, district health officer, district production officer, district veterinary officer) for the supported rendered to us during the study. We also thank our village health teams who guided our research teams into the respective households to conduct the interviews.

## Author Contributions

**Conceptualization:** James Muleme, John C. Ssempebwa, David Musoke, Clovice Kankya, Grace Lubega, Brenda Wagaba, Sonja Hartnack.

**Data curation:** James Muleme, Sonja Hartnack.

**Formal analysis:** James Muleme, Sonja Hartnack.

**Funding acquisition:** James Muleme, John C. Ssempebwa, David Musoke.

**Investigation:** James Muleme, John C. Ssempebwa, David Musoke, Clovice Kankya, Solomon Tsebeni Wafula, Justine Okello, Lesley Rose Ninsiima, Grace Lubega.

**Methodology:** James Muleme, John C. Ssempebwa, David Musoke, Clovice Kankya, Solomon Tsebeni Wafula, Lesley Rose Ninsiima, Rogers Wambi, Sonja Hartnack.

**Project administration:** James Muleme, Solomon Tsebeni Wafula.

**Resources:** James Muleme, John C. Ssempebwa, David Musoke.

**Software:** James Muleme, Solomon Tsebeni Wafula, Sonja Hartnack.

**Supervision:** James Muleme, John C. Ssempebwa, David Musoke, Clovice Kankya, Solomon Tsebeni Wafula, Lesley Rose Ninsiima, Rogers Wambi.

**Validation:** James Muleme, John C. Ssempebwa, David Musoke, Clovice Kankya, Solomon Tsebeni Wafula, Rogers Wambi, Sonja Hartnack.

**Visualization:** James Muleme, Solomon Tsebeni Wafula, Sonja Hartnack.

**Writing – original draft:** James Muleme, Solomon Tsebeni Wafula, Justine Okello, Lesley Rose Ninsiima, Rogers Wambi, James Natweta Baguma, Brenda Wagaba, Sonja Hartnack.

**Writing – review & editing:** James Muleme, John C. Ssempebwa, David Musoke, Clovice Kankya, Justine Okello, Lesley Rose Ninsiima, Rogers Wambi, James Natweta Baguma, Grace Lubega, Brenda Wagaba, Sonja Hartnack.

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
