## [Decision Letter · Decision Letter 0]

7 Dec 2022

PONE-D-22-31194Antimicrobial resistance among farming communities in Wakiso District, Central Uganda: A knowledge, awareness and practice studyPLOS ONE

Dear Dr. Muleme,

Thank you for submitting your manuscript to PLOS ONE. After careful consideration, we feel that it has merit but does not fully meet PLOS ONE’s publication criteria as it currently stands. Therefore, we invite you to submit a revised version of the manuscript that addresses the points raised during the review process. Kindly use the comments to address the specific concerns about your manuscript.

We look forward to receiving your revised manuscript.

Kind regards,

Ismail Ayoade Odetokun, DVM, Ph.D.

Academic Editor

PLOS ONE

https://journals.plos.org/plosone/s/fileid=ba62/PLOSOne_formatting_sample_title_authors_affiliations.pdf.

a) Did participants provide their written or verbal informed consent to participate in this study?

“We would like to thank the farmers in Wakiso district, the district officials (chief administrative officer, district health officer, district production officer, district veterinary officer) for the supported rendered to us during the study. We also thank our village health teams who guided our research teams into the respective households to conduct the interviews. We thank the Makerere University School of Public health small grants scheme for the support in funding this study.”

“The funders had no role in study design, data collection and analysis, decision to publish, or preparation of the manuscript”

Reviewers' comments:

Reviewer's Responses to Questions

**Comments to the Author**

1. Is the manuscript technically sound, and do the data support the conclusions?

Reviewer #1: Yes

Reviewer #2: Partly

2. Has the statistical analysis been performed appropriately and rigorously? 

Reviewer #1: Yes

Reviewer #2: I Don't Know

3. Have the authors made all data underlying the findings in their manuscript fully available?

Reviewer #1: Yes

Reviewer #2: Yes

4. Is the manuscript presented in an intelligible fashion and written in standard English?

Reviewer #1: Yes

Reviewer #2: No

5. Review Comments to the Author

Reviewer #1: This is an interesting study on microbial resistance, which is a rising topic in scientific literature. I have few minor suggestions for the authors:

1. improve English language

2. Add study period to the abstract

3. There is no need to show both yes and no in table 1

4. Table 2 can be presented in manuscript, there is no need for this table

Reviewer #2: This is a very data-rich study conducted in central Uganda with several interesting findings. However, there are also several flaws, some are listed below:

L 83, is there a reference for this very harsh statement?

L 84, why pointing out Makerere like this?

L 109, well described sampling frame, but it would be good to know on which basis the purposively selection was made.

L117, how large was the proportion of farmers that didn’t want to participate?

L121, what is the rational to include ethnic group?

L132-3, What do sources mean, import/export or more immediate sources?

L134, and or? Should be “and, or,”

L144 assistant, plural?

L211 Pls note that antibiotics is a subgroup of antibiotics.

L211-14 can this sentence be reworded? Also, what do microbials refer to?

Table 1, 2, 3 Given the amount of data it is doubtful if these univariate analyses can be justified in the paper in such dense tables. It is suggested that these are put in the annex section. Even if doing so , it is hard for the reader to understand the tables, e.g., in table 1 to which comparison do the OR refer to? What do “none” mean? In Table 2, are the 2 p-values correct? In table 3 seem 0.949% of the farms rear animals and the Non farming type 94.2%. This is confusing for the reader.

L257-60 “or not” reads odd in this sentence. Please reemphasis that several answeres were possible.

L267-268 see above

L269 specify “provider”

L288-289, pls check the wording

L298 Table 4?

Table 4, is the p-value for “income” correct?

Table 5, isn’t there any statistical comparison made here. This very dense table doesn’t give any sensible overview of the results for the reader.

L 352 -359, Pls consider omitting the change to Class1,2,3 -stick instead to the original wording.

L393: Table 6? And no statistical comparisons?

L 424-427, is there a universal scale for level of knowledge -i.e. ,is it valid to compare different studies (ref 24 and 25)?

L 436 (see L 352)

L 440-441 This study was conducted in Sweden, and it might be questioned whether the comparison is at all valid….

L441-3, any reference for this statement.

L443-44 don’t follow the logic in this….

L484-5 can this be better explained?

L498 Would it be worthwhile to mentioned that this was found by two types of analyses Multi regression and PLC?

6. PLOS authors have the option to publish the peer review history of their article (what does this mean?). If published, this will include your full peer review and any attached files.

Reviewer #1: **Yes: **Darko Modun

Reviewer #2: No

---

## [Author Response · Author response to Decision Letter 0]

8 Mar 2023

Response to reviewers comments

https://journals.plos.org/plosone/s/fileid=ba62/PLOSOne_formatting_sample_title_authors_affiliations.pdf.

Authors: These have been revised according to the journal guidelines 

a) Did participants provide their written or verbal informed consent to participate in this study?

Authors: A written informed consent was sought from all the study participants before being recruited into the study (lines 200-201)

Authors: All procedures in the study including consenting were approved under Makerere University School of Public Health Higher Degrees Research and Ethics Committee (SPH-2021-167) and the Uganda National Council for Science and Technology (HS1919ES). – Lines 196 – 198. 

Authors: This has been addressed accordingly in the submission letter. We have removed all funding information from the manuscript. The details we provided in the initial submission still hold. “The funders had no role in study design, data collection and analysis, decision to publish, or preparation of the manuscript.”

“We would like to thank the farmers in Wakiso district, the district officials (chief administrative officer, district health officer, district production officer, district veterinary officer) for the supported rendered to us during the study. We also thank our village health teams who guided our research teams into the respective households to conduct the interviews. We thank the Makerere University School of Public health small grants scheme for the support in funding this study.”

Authors: The sentence on the funder has been removed 

Please remove any funding-related text from the manuscript and let us know how you would like to update your Funding Statement. Currently, your Funding Statement reads as follows: “The funders had no role in study design, data collection and analysis, decision to publish, or preparation of the manuscript”

Authors: This funding statement has been removed from the manuscript. A description has been added in the cover letter for the online form update. 

Authors: This has been addressed accordingly. 

[Note: HTML mark-up is below. Please do not edit.]

Reviewers' comments:

Reviewer's Responses to Questions

Comments to the Author

1. Is the manuscript technically sound, and do the data support the conclusions?

Reviewer #1: Yes

Reviewer #2: Partly

2. Has the statistical analysis been performed appropriately and rigorously? 

Reviewer #1: Yes

Reviewer #2: I Don't Know

3. Have the authors made all data underlying the findings in their manuscript fully available?

Reviewer #1: Yes

Reviewer #2: Yes

4. Is the manuscript presented in an intelligible fashion and written in standard English?

Reviewer #1: Yes

Reviewer #2: No

5. Review Comments to the Author

Reviewer #1: This is an interesting study on microbial resistance, which is a rising topic in scientific literature. I have few minor suggestions for the authors:

Author: Thank you.

1. Improve English language

Author: A thorough English check has been done in the entire manuscript. 

2. Add study period to the abstract

Author: Thank you. This has been added in the abstract and in the methodology section (in between June and September 2021)

3. There is no need to show both yes and no in table 1

Author: We agree. This has been revised accordingly and the No has been deleted. 

4. Table 2 can be presented in manuscript, there is no need for this table

Author: Thank you. We have made table 2 a text within the same position in the manuscript. 

Reviewer #2: This is a very data-rich study conducted in central Uganda with several interesting findings. However, there are also several flaws, some are listed below:

L 83, is there a reference for this very harsh statement?

Author: A clear reference has been added on this statement (Nayiga et al. 2020). The reference states that Wakiso district being a high consumer of antimicrobials than other places in the same study

L 84, why pointing out Makerere like this?

Author: Makerere has been replaced with Uganda. 

L 109, well described sampling frame, but it would be good to know on which basis the purposively selection was made.

Author: A statement qualifying the purposive sampling framework has been added in line 110 to 111 “due to their high involvement in livestock and crop production (District veterinary office records)”.

L117, how large was the proportion of farmers that didn’t want to participate?

Author: The level of rejection or non-response was insignificant. This was because we worked with local leaders, Village health teams and veterinarians serving the areas to explain the study to the participants. These are people trusted by the community and therefore less trouble was faced during data collection. We did not record the proportion of non-respondents. 

L121, what is the rational to include ethnic group?

Author: Ethnic group was important because the study area is predominantly habited by one major group, the Baganda. The other ethnic groups are as well there though in limited numbers. Presumably, their socio-cultural beliefs, norms, and value differ from those of Baganda. It was therefore important to get if there is a difference in knowledge and awareness on Antimicrobials. 

L132-3, What do sources mean, import/export or more immediate sources?

Author: A statement including pharmacy, drug shops, friends, neighbours has been added in the manuscript to mean source

L134, and or? Should be “and, or,”

Author: This has been resolved throughout the document 

L144 assistant, plural?

Author: This has been revised to “assistants”.

L211 Pls note that antibiotics is a subgroup of antibiotics.

Author: This is true. We have revised the statement to mean “other antimicrobials excluding antibiotics”

L211-14 can this sentence be reworded? Also, what do microbials refer to?

Author: This has been revised accordingly 

Table 1, 2, 3 Given the amount of data it is doubtful if these univariate analyses can be justified in the paper in such dense tables. It is suggested that these are put in the annex section. Even if doing so , it is hard for the reader to understand the tables, e.g., in table 1 to which comparison do the OR refer to? What do “none” mean? In Table 2, are the 2 p-values, correct? In table 3 seem 0.949% of the farms rear animals and the Non farming type 94.2%. This is confusing for the reader.

Author: In table 1 we omitted the “no” to make it less dense. Adding one sentence on how to interpret the first OR of females, hint that always the reference category is presented.

For the “none” we specified now, that these are participants which do farming (rearing animals), but have another main source of income. Table 3 renaming “type of farming and “none”) has been done in the revised version. Content of Table 2 is presented in three sentences now. 

L257-60 “or not” reads odd in this sentence. Please reemphasis that several answers were possible.

Author: This has been revised accordingly. 

L267-268 see above

Author: This has been addressed as above 

L269 specify “provider”

Author: This has been added as drug shop attendant, veterinarians, friends, neighbours etc

L288-289, pls check the wording

Author: This has been revised accordingly. 

L298 Table 4?

Author: All the nomenclature of tables has been revised accordingly 

Table 4, is the p-value for “income” correct?

Author: We checked it and it was correct 

Table 5 isn’t there any statistical comparison made here. This very dense table doesn’t give any sensible overview of the results for the reader.

Author: We agree that table 5 and 6 are very dense. The reason why we would prefer (but are willing to change) is that this table presents the proportions of the different answer levels (correct, wrong, or unknown) cross-classified by the three different categories in our sample. In other words, based on this table, the reader could a) assess the results of the polytomous latent class analysis and b) assess if individuals questions raised a different proportion in correct answers. We agree that there is no explicit statistical comparison made – still if the 9%% CI are not overlapping, then one could conclude that there is a statistical significant difference (alpha = 0.05, see the ASA statement on p-values). 

L 352 -359, Pls consider omitting the change to Class1,2,3 -stick instead to the original wording.

Author: We are not sure about this comment. The designation class1 to 3 results from the polytomous latent class analyses. We looked at these classes and the proportions of correct/wrong or unknown answers, and based on this the names of “appropriate knowledge” etc. If the reviewer prefers, we could omit this naming and just speak of class 1, 2 or 3.

L393: Table 6? And no statistical comparisons?

Author: (please see above our answer to table 5)

L 424-427, is there a universal scale for level of knowledge -i.e., is it valid to compare different studies (ref 24 and 25)?

Author: Thank you, we agree. A statement “There is need for researchers and internationally recognized research bodies to generate a knowledge measurement scale to enable a proper comparison of knowledge across different studies and settings” has been added. 

L 436 (see L 352)

Author: If the reviewer prefers, we could mention here too class 1 “appropriate knowledge”. Since the designation of an individual belonging to a specific class is the main outcome of the polytomous latent class, we think it not appropriate to omit “class 1”.

L 440-441 This study was conducted in Sweden, and it might be questioned whether the comparison is at all valid….

Author: This has been changed 

L441-3, any reference for this statement.

Author: The reference has been changed from Vallin to Sandato. The statement following has been removed. 

L443-44 don’t follow the logic in this….

Author: This has been revised and the unclear statement deleted accordingly. 

L484-5 can this be better explained?

Author: This sentence has been rephrased accordingly 

L498 Would it be worthwhile to mentioned that this was found by two types of analyses Multi regression and PLC?

Author: Thank you, this is a good addition. We have made the necessary changes accordingly

---

## [Decision Letter · Decision Letter 1]

10 Apr 2023

Antimicrobial resistance among farming communities in Wakiso District, Central Uganda: A knowledge, awareness and practice study

PONE-D-22-31194R1

Dear Dr. Muleme,

We’re pleased to inform you that your manuscript has been judged scientifically suitable for publication and will be formally accepted for publication once it meets all outstanding technical requirements.

Kind regards,

Ismail Ayoade Odetokun, DVM, Ph.D.

Academic Editor

PLOS ONE

Additional Editor Comments (optional):

Reviewers' comments:

Reviewer's Responses to Questions

**Comments to the Author**

1. If the authors have adequately addressed your comments raised in a previous round of review and you feel that this manuscript is now acceptable for publication, you may indicate that here to bypass the “Comments to the Author” section, enter your conflict of interest statement in the “Confidential to Editor” section, and submit your "Accept" recommendation.

Reviewer #1: All comments have been addressed

Reviewer #3: All comments have been addressed

2. Is the manuscript technically sound, and do the data support the conclusions?

Reviewer #1: Yes

Reviewer #3: Yes

3. Has the statistical analysis been performed appropriately and rigorously? 

Reviewer #1: Yes

Reviewer #3: Yes

4. Have the authors made all data underlying the findings in their manuscript fully available?

Reviewer #1: Yes

Reviewer #3: Yes

5. Is the manuscript presented in an intelligible fashion and written in standard English?

Reviewer #1: Yes

Reviewer #3: Yes

6. Review Comments to the Author

Reviewer #1: The manuscript has been changed according to the Reviewers' comments. I suggest to accept the manuscript.

Reviewer #3: (No Response)

7. PLOS authors have the option to publish the peer review history of their article (what does this mean?). If published, this will include your full peer review and any attached files.

Reviewer #1: No

Reviewer #3: No

---

## [Editor Report · Acceptance letter]

13 Apr 2023

PONE-D-22-31194R1 

Antimicrobial resistance among farming communities in Wakiso District, Central Uganda: A knowledge, awareness and practice study 

Dear Dr. Muleme:

I'm pleased to inform you that your manuscript has been deemed suitable for publication in PLOS ONE. Congratulations! Your manuscript is now with our production department. 

Kind regards, 

on behalf of

Dr. Ismail Ayoade Odetokun 

Academic Editor

PLOS ONE